# Integrating Policy Summaries with Reward Decomposition Explanations

## Abstract

Explaining the behavior of agents operating in sequential decision-making settings is challenging, as their behavior is affected by a dynamic environment and delayed reward. In this paper, we study a new way of combining local and global explanations of sequential decision-making agents in order to help understand their behavior. Specifically, we combine reward decomposition, a local explanation method that exposes agent preferences, with HIGHLIGHTS, a global explanation method that shows a summary of the agent's behavior in "important" states. We conducted a user study to evaluate the integration of these explanation methods and their respective benefits. Our results show that local information in the form of reward decomposition contributed to participants' understanding of agents' preferences, while HIGHLIGHTS summaries did not lead to an improvement compared to a baseline showing frequent agent trajectories.

## Introduction

Artificial Intelligence (AI) agents are being deployed in a variety of fields such as self- driving cars, medical care, home assistance and more. As this field develops the need of understanding this agents behavior has become clear.

In this work, we focus on explaining the behavior of agents that operate in sequential decision-making settings, which are trained in a deep reinforcement learning (RL) framework. We study the effectiveness of providing users with global and local explanations on the behavior of RL agents. Global explanations explain the general behavior of the model, e.g., by describing decision rules or strategies. In contrast, local explanations try to explain specific decisions that an agent makes. In this paper, we will explore the combination of global and local explanations. Specifically, we focus on policy summaries as a global explanation method, and reward decomposition as the local explanation method. Policy summaries aim to convey the strategy of the agent by demonstrating its behavior in a selected set of world states (Amir and Amir 2018). Reward decomposition aims to reveal the agent's reasoning in particular decision points by decomposing the rewards to sub-rewards (e.g., a reward for driving in the right lane, a reward for driving fast, etc.), al-

lowing to compare the tradeoffs between different actions with respect to reward types (Juozapaitis et al. 2019).

We conducted an experiment in which participants were randomly assigned to one of four different combinations of (1) having or not having a local explanation (reward decomposition) and (2) having a global (strategy summarization HIGHLIGHTS (Amir, Doshi-Velez, and Sarne 2019)) explanations or having random states chosen (Table 3). We used a highway domain, where the RL agent controls a self driving car. On our main task the participants' needed to determine the agents' preference for different agents they saw. We also asked the participants to rate their confidence in their answers and we measured their satisfaction using Hoffman's explanation satisfaction scale (Hoffman et al. 2018).

Our results show that the use of reward decomposition as a local explanation helps users comprehend the agents' preferences. Moreover, The different explanation methods did not result in differences in terms of participants' confidence or satisfaction even though understanding graphs might be more difficult then video summarization.

## Related Work

In this section, we review related work from two areas related to this article: (1) explanations for reinforcement learning agents, and (2) explanations for self-driving cars.

### Explainable Reinforcement Learning

Explainable reinforcement learning methods generate explanations for agents and policies in sequential decision making scenarios. Broadly, there are two classes of explanations: *local* explanations which explain why a particular action was taken, for example (Madumal et al. 2020; Tabrez, Agrawal, and Hayes 2019; Hilton et al. 2020; Dodson, Mattei, and Goldsmith 2011). In many RL methods the Q -value is used in order to select actions. However, it does not supply information with respect to the factors that contribute to the action choice. Juozapaitis et al. (Juozapaitis et al. 2019) used reward decomposition to get insights on these factors. Decomposing the environment's reward into a sum of meaningful reward types enables to provide explanations on which action has an 'advantage' over other actions. As well as this paper we also use reward decomposition as a mean of providing explanations.

Anderson et al. (Anderson et al. 2019) present a user study that investigate the impact of explanations on non-experts' understanding of reinforcement learning agents. They investigate both a common RL visualization, saliency maps, and reward decomposition bars. They designed a four-treatment experiment to compare participants' mental models of an RL agent in a simple Real-Time Strategy game. Their results show that the combination of both saliency and reward bars were needed to achieve a statistically significant improvement in mental model score over the control.

The other class of explanation is *global* explanations which attempt to describe the high-level policy of the agent (Booth, Muise, and Shah 2019). An example for global explanations is strategy summarization (Amir, Doshi-Velez, and Sarne 2019) which demonstrates an agent's behavior in carefully selected world states. The states can be selected based on different criteria, e.g., state importance (Amir and Amir 2018) or using machine teaching approaches (Lage et al. 2019).

The *combination of local and global explanations* has been studied prior. Similarly to our work, Huber et al. (Huber et al. 2020) combined local and global explanation methods in RL agents. They used strategy summaries (global explanation) with saliency maps (local explanation). Since this study showed that using saliency maps as local explanations is lacking, we study the integration of reward decomposition as local explanations, together with policy summaries.

### Explaining in the domain of self driving cars

In recent years, the importance of explainability in the field of self driving cars is receiving more attention. Wiegand et al. (Wiegand et al. 2019; Koo et al. 2015) used visualizations to explain self-driving vehicle behavior. Shen et al. (Shen et al. 2021) created a framework which aims to help users of autonomous vehicles preview autopilot behaviors of updated control policies prior to purchase or deployment in order in order for the users to trust the self-driving technology. Our work is not specific to the driving domain, but can also be applied to it, as done in our user study.

## Background

In this section, we review the technical background underlying the current work. We first describe reinforcement learning and deep reinforcement learning. We then describe the two explanation methods that this work integrates: reward decomposition and policy summaries.

### Reinforcement Learning

Reinforcement learning is a computational approach to understanding and automating goal - directed learning and decision making (Sutton and Barto 2018). Reinforcement learning uses a formal framework defining the interaction between a learning agent and its environment in terms of states, actions, and rewards. This is done by estimating a Q-value function.

We assume a Markov Decision Process (MDP) setting. Formally, MDP is a tuple $< S, A, R_a, Tr >$:

- $S$: Set of states.

- $A$: Set of actions.
- $R_a$: The reward received after transitioning from state s to state $s'$, due to action $a$.
- $Tr$: A transition probability function $Tr(s, a, s') \rightarrow [0, 1] s.t s, s' \in S, a \in A$ defining the probability of transitioning to state $s'$ after taking action $a$ in $s$.

An agent's policy $\pi(s, a)$ is a probability distribution over the set of possible actions in a given state. The Q-function is defined as the expected value of taking action $a$ in state $s$ under policy $\pi$ throughout an infinite-horizon while using a discount factor $\gamma$.
$Q^\pi(s, a) =^\pi [\sum_{t=0}^{\inf} \gamma^t R_{t+1} | s_t = s, a_t = a]$. $Q^*(s, a)$ denotes the Q-function of the optimal policy $\pi^*$ meaning, $\pi^*(s) = argmax_{a \in A} Q^*(s, a)$.

**Deep Reinforcement Learning** Deep reinforcement learning uses deep neural networks to approximate any of the components of reinforcement learning such as the Q-value, policy and model (state transition function and reward function).

One common approach is using a deep Q network (DQN). DQN is a multi-layered neural network that for a given state $s$ and action $a$ outputs a vector of action values $Q(s, a; \theta)$, where $\theta$ are the parameters of the network. For an $n$-dimensional state space and an action space containing $m$ actions, the neural network is a function from $R^n$ to $R^m$. The DQN contains two networks, the target network and the value network. The target network, with parameters $\theta^-$, is the same as the value network except that its parameters are copied every $\tau$ steps from the value network i.e. $\theta_t^- = \theta_t$ and kept fixed on all other steps. The target used by DQN is, $Y_t^{DQN} \equiv R_{t+1} + \gamma max a Q(S_{t+1}, a; \theta_t^-)$. The estimation of $Q$ is done by minimizing the sequence of loss functions: $L_i = E_{s,a,r,s'}[(y_i^{DQN} - Q(s, a; \theta_i))^2]$ Moreover, there is an experience replay, observed transitions are stored for some time and sampled uniformly from this memory bank to update the network.

In this work, we use the Double Deep Q Network architecture. In Double DQN we replace the target $Y_t^{DQN}$ with $Y_t^{DoubleDQN} \equiv R_{t+1} + \gamma Q(S_{t+1}, argmax a Q(S_{t+1}, a; \theta_t), \theta_t^-)$. The update to the target network stays unchanged from DQN, and remains a periodic copy of the value network (Van Hasselt, Guez, and Silver 2016).

### Reward Decomposition

Originally, reward decomposition was used in order to expedite the learning rather than giving explanations. Van Seijen et al. (Van Seijen et al. 2017) proposed the Heirarchical Reward Architecture (HRA) model. HRA takes as input a decomposed reward function and learns a separate value function for each component reward function. Because each component typically only depends on a subset of all features, the corresponding value function can be approximated more easily by a low-dimensional representation, enabling more effective learning.

Prior work has suggested the use of reward decomposition as a local explanation. Raw Q - values do not give any

insight into the positive and negative factors contributing to the preferences since the individual reward types are mixed into a single reward scalar. We can explicitly expose the different types of rewards to the agent via *reward decomposition* (Juozapaitis et al. 2019).

The MDP formulation can incorporate reward decomposition by specifying a set of reward components $C$ and defining a vector-valued reward function $\rightarrow R : S \text{ x } A \rightarrow R^{|C|}$, where $R_c(s, a)$ is the reward for component $c \in C$. The objective remains the same, to optimize the overall reward function $R(s, a) = \sum_{c \in C} R_c(s, a)$. However, we can define vector-valued Q-function $\rightarrow Q^\pi$, where $Q_c^\pi(s, a)$ gives action values that account only for rewards related to component $c$. As a result of these definitions, the overall Q-function also decomposes since $Q^\pi(s, a) = \sum_c Q_c^\pi(s, a)$.

## Policy Summaries

*"Agent strategy summarization"* (Amir, Doshi-Velez, and Sarne 2019) is a paradigm for conveying the global behavior of an agent. In this paradigm, the agent's policy is articulated by a carefully selected set of world states that conveying the agent's behavior. The goal in strategy summarization is choosing the subset of state-action pairs that best describes the agents policy. In a formal way, Amir & Amir (Amir and Amir 2018) defined the set $T = < t_1, ..., t_k >$ as the trajectories that are included in the summary, where each trajectory is composed of a sequence of $l$ consecutive states and the actions taken in those states, $< (s_i, s_i), ..., (s_{i+l-1}, a_{i+l-1}) >$. Since it is not feasible for people to review the behavior of an agent in all possible states, $k$ is defined as the size of the summary e.g $|T| = k$.

In our work we use a summarization approach called HIGHLIGHTS (Amir, Doshi-Velez, and Sarne 2019) that extracts "important" states from execution traces of the agent. An important state is denoted as $I(s)$ and is defined as: $I(s) = max_a Q_{(s,a)}^\pi - min_a Q_{(s,a)}^\pi$. According to this formulation, a state is considered important if there is a large gap between the expected outcome of the best and worst action available to the agent in the state. For example, a car reaches a crossroads, choosing one path will lead the car to a congested road while the other path will lead to a quick arrival. The crossroads will be an important state since the action chosen in that state will have a significant impact on arrival time.

# Integrating Policy Summaries and Reward Decomposition

In this section, we describe the framework and the architecture used in this study to integrate global and local explanations.

## Neural Network Architecture

We decomposed the reward function $R$ into $|C|$ reward functions: $R(s, a, s') = \sum_{c=1}^{|C|} R_c(s, a, s') \forall s, a, s'$, where in our study we decided to set $|C| = 3$ for the basic case that each reward function has only one reward type that is positive (right lane, change lane and high speed). Since each sub-reward $c$ has a different policy, it has also its own Q-value

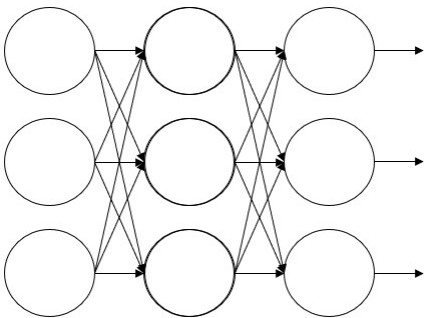

Figure 1: Neural network with three heads

function, $Q_c$. In general, different policies can share multiple lower-level layers of a Double DQN. Therefore, the combined weights of the agents can be describe by a single vector $\theta$. The combined network that represents all Q-value functions is called the Hybrid Reward Architecture (HRA) (Van Seijen et al. 2017). The actions that are selected for the HRA are based on the sum of the agent's Q-value functions: $Q_{HRA}(S, A; \theta) := \sum_{c=1}^{|C|} Q_c(s, a; \theta) \forall s, a$

Alternatively, the collection of agents that have one type of reward can be viewed as a single agent with multiple *heads*, such that each head calculates the action-values of a current state under his reward function. Since the sum of all heads should be equivalent to the original neural network there are small adjustments that needed to be made, e.g., the sum of the normalization of each head should be equal to the normalization of the original neural network. Each head calculates it's loss function i.e. $L_i(\theta_i) = E_{s,a,r,s'}[\sum_{c=1}^{|C|} (y_{c,i}^{DoubleDQN} - Q_c(s, a; \theta_i))^2]$.

We decomposed the reward function into three different reward functions, one per each reward type. The networks input is an array of size 25 (5X5) that represents the state. This is followed by two fully connected hidden layer of length 256. The last layer is connected to three heads (Figure 1). Each head consists of a linear layer and outputs a Q-value array in length of 5 that contains the following: lane left, idle, lane right, faster, slower. In addition, each head computes a loss function by bellman residual and the loss function of the neural network is the sum of all three.

## Integrating HIGHLIGHTS with Reward Decomposition

We combine HIGHLIGHTS as a global explanation with reward decomposition as a local explanation. We used HIGHLIGHTS to find the most important states in each episode. For each state that was chosen, we created the reward decomposition bars that depict the Q-values for each action in the chose state.

## Sanity Check

We first wanted to ensure that the changes made to the neural network architecture did not create an inherent problem and resulted in learning comparable to that of an agent trained without decomposed rewards. To this end, we compared the

cumulative rewards of two RL agents that differ only by their neural network structure and the normalization. Table 1 presents the main parameters that where used for each RL agent. The results show that the average reward of a RL agent with multiple heads is in the same range as a RL agent that had an original neural network (see Table 1). We can conclude from this check that in our domain using reward decomposition did not harm the performance of the agent.

## Empirical Methodology

We conducted a user study in which participants were shown videos or images of four different agents. For each agent, they were asked to rank the reward of different actions the agent can take being that the reward preference reflects the agents strategy.

### Experimental domain and agent training

We used a multi-lane highway environment (as seen in the top part of Figure 2) for our experiments. The environment allows us to control different variables such as the amount of vehicles, vehicles density, rewards, speed range and more.

In the environment, the RL agent - a self driving car- is trained using a double DQN architecture. The model estimates the state-action value function and produces a greedy optimal policy. The objective of the agent is to maximize its reward by navigating a multi-lane highway that includes other vehicles. Positive rewards can be given for each of the following actions: changing lanes (CL), speeding up (SU) and moving to the right most lane (RML). We trained four RL agents in this domain which differ in their policies.

1. The Good Citizen - Highest reward for being in the right lane, next to change lane and lastly to speed up.

2. Fast And Furious - Highest reward for speeding up, then to change lanes and lastly to be in the right most lane.

3. Dazed and Confused - Highest reward for changing lanes, next to be in the right most lane and lastly to speed up.

4. Basic - Reward for being in the right most lane.

Common to all agents, when crashing a negative reward of -3 is given, and no future rewards can be obtained due to ending the episode. The settings of the rewards for the different agents that we used are summarized in Table 2.

Each agent was trained for 2,000 episodes and each episode included 80 time stamps (or fewer if the agent crashed). In each time stamp the agent makes a decision, i.e., an action. The product of each episode is a video that describes the agent's behavior (trace of actions).

Our implementation is based on an open source highway environment and RL agents[1].

### Study Design

*Empirical domain*. We used a highway simulator environment (as seen in the upper part of Figure 2) for our experiments. We chose this environment for two reasons. First,

_______________
[1]https://github.com/eleurent/highway-env, https://github.com/eleurent/rl-agents

basic traffic laws are known to most adults and therefore no additional domain knowledge was necessary to understand using this domain. Second, it is easy to train agents that differ qualitatively in their behavior by modifying the reward function.

*Experimental conditions*. In order to evaluate the impact of combining global and local explanations, as well as each type individually, we assigned participants to one of four different conditions (as shown in Table 3). We set $k = 8$, the size of the summary we compute using HIGHLIGHTS therefore, all participants were shown a summary of the agents behavior that is composed out of 8 different videos or images regarding the specific agent. More specifically, participants that were in conditions including RD method received images while the other received videos. The summaries were chosen through different methods as follows:

- Frequency sampling summaries (FS): In this condition, we randomly selected states from the simulations of the agent. Since each state has the same probability of being chosen, in practice states that appear more frequently had more chances of being selected and are more likely to appear in the summary. Therefore, this is equivalent to selecting states based on the likelihood of their appearance. Moreover, to ensure that the summary is not particularly good or particularly bad we created 10 different summaries of this form.

- Highlight Summaries (H): In this condition, participants were shown summaries that were generated by the HIGHLIGHTS algorithm.

- Frequency sampling summaries + Reward decomposition (FS+RD): In this condition, participants were shown images of states that were generated by the likelihood based summaries. Each chosen state was shown using an image along with a reward decomposition bar plot that represents the Q-values of the different components for each available action in the chosen state, as shown in Figure 2 .

- Highlight + Reward decomposition (H+RD): In this condition, participants were shown images of states that were selected by the HIGHLIGHTS algorithm. However, since in this condition participants are shown images, they only see the most "important" state, meaning that they did not get the context to that state as the HIGHLIGHT algorithm provides. These are the same states that appeared in the H summaries. For each state (Figure 2), a reward decomposition bar that matches the state depicted in the image was shown along with the image.

After training each agent for 2,000 episodes we then created the same amount of simulations that we stored videos of traces from which we sampled

*Procedure*. At first, participants were given an explanation regarding the domain of the experiment. Second, they were given a brief explanation about reinforcement learning and specifically about q-value (the explanation were given in layperson vocabulary and did not go into details). Lastly, participants where given information about the type of explanation they will see in the survey and an example. In

| | Original | Multi heads |
|---|---|---|
| Type | Multi Layer Perceptron | Multi Layer Perceptron |
| Method | Epsilon Greedy | Epsilon Greedy |
| Loss function | L2 | L2 |
| Duration of each episode | 40 time stamps | 40 time stamps |
| Number of lanes | 4 | 4 |
| Number of vehicles | 30 | 30 |
| | right lane=5 | head 1 right lane=5
head 1 high speed=0
head 1 lane change=0 |
| Reward | high speed=5 | head 2 right lane=0
head 2 high speed=5
head 2 lane change=0 |
| | lane change=5 | head 3 right lane=0
head 3 high speed=0
head 3 lane change=5 |
| Normalization | [0,1] | [0,1/3]-each head |
| Number of episodes | 2000 | 2000 |
| Average result of reward | 38 | 39 |

Table 1: Main values and results of RL agent with original neural network vs. multi head neural network

| | CL
reward | SU
reward | RML
reward |
|---|---|---|---|
| The Good Citizen | 3 | 1 | 8 |
| Fast and Furious | 5 | 8 | 1 |
| Dazed and Confuse | 8 | 1 | 5 |
| Basic | 0 | 0 | 15 |

Table 2: The settings of the four agents.

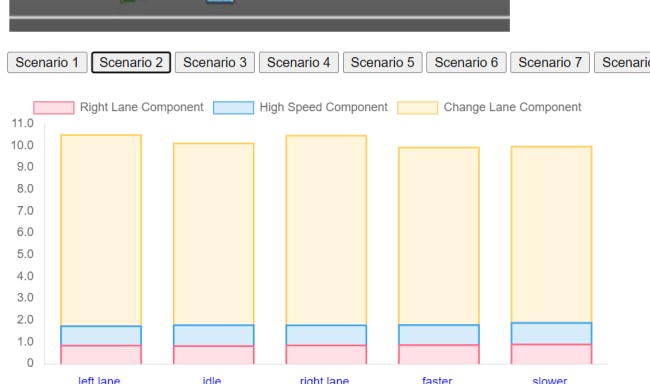

Figure 2: The mean success of all agents by conditions

order to keep the conditions as similar as possible, all participants were given the same amount of information and knowledge prior to beginning the survey tasks. Moreover, at the end of each instructions phase the participants were asked to complete a quiz, and were only allowed to proceed with the survey after answering all questions correctly. Participants were compensated as follows: they received a $3 base payment, and an additional bonus of 10 cents for each correct answer.

*Task.* The participants' task was to assess the preferences of four different agents (self-driving cars) based on the provided explanations of the agents' behavior. The ordering of the agents was random. Specifically, participants where asked to rank which of a pair of options (e.g., high speed vs. driving in the right lane) the agent prioritizes or whether it is indifferent between the options. This was done for each pair of reward components.

- high speed vs. driving in the right lane
- driving in the right lane vs. changing lanes
- changing lanes vs. high speed

Participants were then asked to rate their confidence in each of their answers regarding the agents' preferences on a scale from 1 to 5 where 1 is "not confident at all" and 5 is "very confident". Lastly, participants completed several explanation satisfaction questions adapted from the question-

naire proposed by (Hoffman et al. 2018). The questions are the following:

1. The videos\ images helped me to recognize agent behaviours
2. The videos\ images contain sufficient detail for recognizing agent behaviours
3. The videos\ images contain irrelevant details
4. The videos\ images were useful for the task
5. The specific scenarios shown in the videos\ images were useful for the task.

|        | FS summaries | Highlights |
|--------|-------------|-----------|
| No RD  | FS          | H         |
| RD     | FS+RD       | H+RD      |

Table 3: The four study conditions.

*Participants*. We recruited participants through Amazon Mechnical Turk (N = 164). We excluded participants who did not answer the attention question correctly, as well as participants who completed the survey in less than 7 minutes or in less than two standard deviations from the mean completion time in their condition.

After screening, we had 127 participants (mean age = 36 years, 58 female, all from the US, UK, or Canada). Participants were randomly assigned to one of the four conditions.

## Results

We found that reward decomposition significantly improved participants' ability to asses the agents' preferences, as shown in Figure 3. When breaking down correctness rates by agents, we find that RD was beneficial for assessing all four agents (see Figure 4). We report the mean values and the 95% confidence interval (CI) computed using the bootstrap method. In all plots the error bars correspond to the 95% confidence intervals.

In addition, our results show that the combination of H+RD helped asses the agents preferences when the difference between the reward types was minor. For example, when assessing the agent "Fast and Furious", that was trained according to the rewards of 8 points for speed up vs. 5 points for changing lanes, participants who where shown H+RD succeeded 79% of the times compared to participants in conditions FS+RD, FS or H that succeeded 61%, 8% and 14% respectively. This indicates that even though our overall results do not show that the combination of H+RD is significantly better there were cases in which this combination significantly helped.

We did not find differences between conditions with respect to participants' confidence. We checked the mean confidence rating the participants assigned for each agent in a scale of 1 – 5. As seen in Figure 5, for every condition the confidence of the participants is above neutral rating. When assessing participants' satisfaction, we also did not find differences between the conditions. Thus, while participants' objective performance was better with RD compared to video-based policy summaries, this did not lead to an increase in the subjective measures.

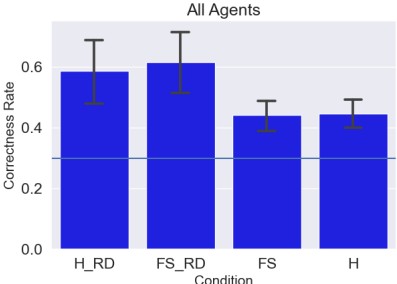

Figure 3: The mean success of all agents by conditions

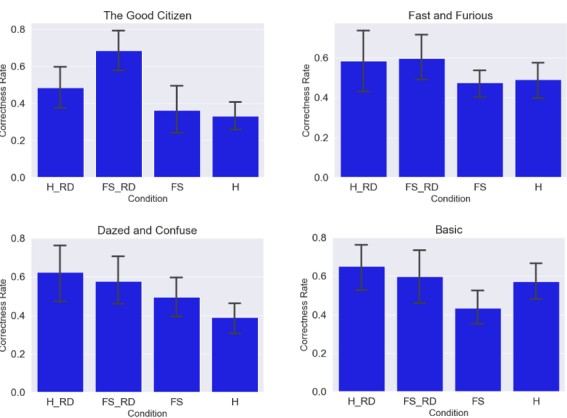

Figure 4: The mean success for each agent by conditions

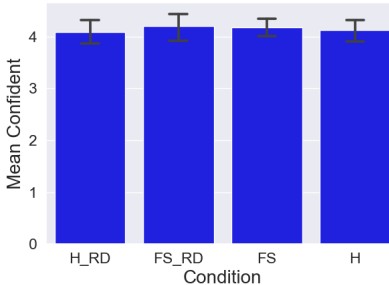

Figure 5: The mean confidence for all agents by conditions

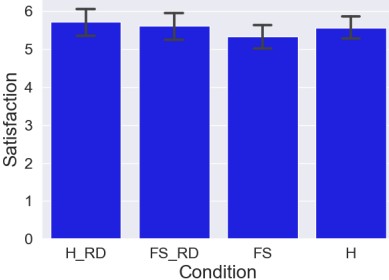

Figure 6: The mean satisfaction from explanations by conditions

## Discussion and Future Work

With the progress of AI, circumstances which require people to understand and trust AI agents are more likely to appear. This paper presented a new approach for understanding RL agents. Experimental results show that RD helps users understand the agents' decision making policy, specifically, it helped users analyze the agents' preferences.

Our results do not align with the results of prior work (Huber et al. 2020). Huber et al. found that HIGHLIGHTS worked and adding saliency maps as a local explanation did not add much, while in our study local explanations (RD) where more important than the global explanation. Our hypothesis as to why HIGHLIGHTS did not contribute to the success of the participants is since we set $k$ to high. The domain that we choose is limited in the number of different scenarios we can simulate. Therefore, when setting the size of the summary to be to high, even without using HIGHLIGHTS the participants were shown enough scenarios that gave a good enough picture of the agents. Another hypothesis is that HIGHLIGHTS was shown to work when participants' were asked to compere between different agents while in our study we asked participants' to comer between different actions of the same agent

As for future work, we note the following possible directions: i) testing whether RD can help participants predict the agents' actions in unseen states; ii) evaluating the explanations in additional domains; iii) creating a better visualization for RD that can contain the same data but in a more accessible way.

## Acknowledgments

To Yotam Amitai for helpful discussions.

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

## Appendix

Survey link: https://technioniit.eu.qualtrics.com/jfe/form/SV_6yQGU7HV8p3RiwC