# OpenReview forum: "Integrating Policy Summaries with Reward Decomposition Explanations"
_icaps-conference.org/ICAPS/2022/Workshop/XAIP — XAIP 2022_

### Official Review · Reviewer_LMjC · 2022-04-27
**This paper in its current version has many issues in its presentation, detail, and evaluations**

**Rating:** 5
**Confidence:** 4

**Review:**

Contribution and Summary:
The main contribution of this paper is that they combined reward decomposition explanation (local explanation) and an existing policy summarization method called HIGHLIGHT (global explanation). They did a user study on a self-driving car domain (multi-lane highway environment) to evaluate whether reward decomposition explanation combined with global explanation will decrease the users’ error in understanding the agent’s preferences. They compared the combined explanation with baselines where there is no reward decomposition explanation added. They also evaluate users’ confidence and satisfaction through a subjective questionnaire to see how different explanations (their combined explanation vs. baselines) would affect these factors.

The takeaways of this work are that combining reward decomposition explanation with global explanation will increase the correctness rate of the users, but it will not have any effects on users’ confidence and satisfaction.

Major Comments:
In my opinion, while a good presentation and the right evaluation of the idea could make the work an interesting work, this paper in its current version has major issues in its presentation, detail, and evaluations. The main motivation of this paper is not clear as to why such explanations are needed and what would this work and its result add to the community; the evaluations also are incomplete and the result cannot be concluded in this current state. My comments are as follows.

1. The major motivation of the paper is not clear; why such integration and study is needed?
2. In the related work, more detail needs to be provided about the cited papers. For instance, a sequence of references is listed for the local explanation, but some detail about each of these works should be provided (what are their methods and takeaways). This is the case for the cited papers on explainability in self-driving cars. More importantly, after that, you need to provide the motivation and differences of these papers and how your paper stands out from the related works.
3. On page 2, it is written that the work uses the Double DQN, but without explaining why and the reason for using double DQN instead of alternatives. More details need to be provided.
4. Figure 1 is very premature where it is just showing a neural network with three heads. What exactly did you want to present with this image? I think anybody knows what a three heads NN looks like. Instead, it’s good to have a more informative figure that showed the architecture used in the paper.
5. In the section Integrating HIGHLIGHT with reward decomposition, it’s not clear what is the change you made in the neural network and what are the differences. It’s useful to have a figure to show both architectures, and more detail is needed to explain both clearly and explain the reason for having this multi-head NN. Furthermore, the integration part is also not clear, how do you integrate HIGHLIGHT and reward decomposition and what does it mean by their integration?
6. It’s not clear why the multi-head is needed at all and what its use and benefit of it? Is it for integration or something else. Major detail is needed in the integration section.
7. The paper wasn't organized well, especially in empirical evaluation. For example, it first talked about the 4 conditions of the study, and the table then gave the detail, while the detail should come before that. This is the case when talking about the image and video.
8. In FS+RD, it should be clarified how the states are shown to the users. I could later guess based on the figure but more detail needs to be provided.
9. On H+RD page 4, it is written “meaning that they did not get the context to that state as the HIGHLIGHT algorithm provides”. What is the context that is missing here?
10. In FS, it is not clear how many states are selected?
11. FS is another way of policy summarization, so it’s better to have an additional baseline without any explanation to have a useful comparison.
12. In the evaluation, the users get to assess the preferences in a pair-wise reward component. While the paired comparison is useful, in my opinion, all 4 also should have been rated besides each other to get the right evaluation.
13. In table 2, how and why these rewards are chosen? What is the base for these choices and how the change in the rewards will affect the results?
14. The paper evaluates the correctness rate of the users in knowing the agent’s preferences, in my opinion, this is a factor influenced by many things such as the users’ attention, and level of understanding of the explanation. So, to evaluate this, you should measure their attention and their understanding as well to see how much the result is dependent on the provided explanation as other dependent variables. Moreover, it is important to make this clear that the user’s correctness rate is different than their understanding of the explanation (something that is not evaluated here)
15. The paper excludes participants who didn’t answer attention questions and didn’t complete the survey in less than 7 min. What is the attention question? What is the basis for choosing the 7 min criterion for excluding participants?
16. The results and evaluation are incomplete and inconclusive, you can’t say one condition is significantly better than the other without actually evaluating them with a proper statistical test and without calculating the p-value. The paper needs to provide clear hypotheses and then show them by statistical tests.
17. It was written that “In addition, our results show that the combination of H+RD helped assess the agents preferences when the difference between the reward types was minor” why does it matter? I think just having a higher correctness rate for H+RD is not enough to conclude that. Not only should a statistical test be done to evaluate this but also in my opinion many different factors might influence the condition which makes it hard to conclude this even with a statistical test.
18. Similar to previous points, the claim that H+RD is significantly better and there were cases in which this combination significantly helped is questionable without actual evaluation.

Minor Comments:

1. Some of the equations are written in a bad format such as missing space and combining text and math which is confusing. For example, on page 2, [T r(s, a, s′ ) → [0, 1]s.ts, s′ ∈ S, a ∈ A] there should be space between different elements here. [ Rn to Rm] should be [Rn → Rm]. Also, some notations are very confusing to me (doesn’t look right) and I am not sure if this is the standard way of writing them. Examples [a vector-valued reward function →R : S x A → R|C|] and [ vector-valued Q-function →Qπ]. It’s not clear what is the meaning of the first arrow beside the text?
2. On page 3, it’s not clear if the lane left, idle, lane right, faster, slower are the actions or something else?
3. Change lane, speeding up, and moving to the right lane were introduced as components of the reward function, and then later on page 4, they are mentioned as actions. Please clarify this.

Typos:

There are many typos in the paper, here are some of them:
1. Page 1, [understanding graphs might be more difficult then video summarization.] → [than]
2. Page 2, [in order in order for the users] ??
3. Page 2, [Li = Es,a,r,s′ [(y DQN i −Q(s, a; θi))2 ]] → missing ‘.’ after equation.
4. Page 3, [ < (si , si), ...,(si+l−1, ai+l−1) >.], second si should be ai.
5. Page 3, [it’s loss function] → its loss function
6. Page 6, [the agents preferences] → agent’s preferences
7. Page 7, [participants’ were asked to compere between different agents] → compare
8. Page 7, [we asked participants’ to comer between different actions] → compare


Suggestions and Questions to the Authors:
My suggestions and questions to the authors are what I mentioned in the comments.

---

### Official Review · Reviewer_uUHq · 2022-04-30
**Review for paper id 5**

**Rating:** 6
**Confidence:** 4

**Review:**

The paper looks at the problem of combining local and global explanations to provide more effective explanations. Specifically, they consider the use of HIGHLIGHTS which provides policy summaries in the form of example state, action pairs (chosen based on specific selection criteria), and the local explanation takes the form of reward component decomposition of the Q values. The approach was evaluated using a user-study on a simulated driving domain, where the participants were shown sample agent behavior (in some conditions chosen using HIGHLIGHTS algorithm) with possible local explanations and then asked questions regarding the agent characteristics.

On the positive side, I believe the paper looks at a very interesting problem. The need to provide explanation at different levels of fidelity is an important problem, and I particularly like the idea of enriching summaries with local explanation. I also appreciate the fact that the authors actually ran user studies to evaluate the approach.

However I do have some reservations about the paper, in particular regarding two main aspects

Choice of Global and Local Explanation - Why were these particular global and local explanation techniques selected? For local-explanation, while there exists so many possible choices, was there any particular characteristics of this local explanation that made it seem like an interesting choice as local explanation? Ideally you would want the local explanation to feed into the specific global explanation being used an vice versa. Could you have use clear distinction between the different reward components as a possible way to choose the high level states to be shown as part of the summary. Currently, I don't see any reason to believe another local explanation wouldn't have worked as well or better in this context. If in fact, the authors goal is to try to find such combination of global and local explanation through user studies, I would recommend doing a larger study where various global and local explanations are tried at the same time.

Details Regarding the Technical Approach: Unfortunately, current writing of the paper makes it a bit hard to properly evaluate many of the technical aspects of the paper. The problems here include issues from inconsistant notation usage to a lot of important details being missing. Just to cite an example, I am assuming the symbols Y_t and y_t are being used interchangeably.  More importantly in the section 'Neural Network Architecture', I was actually quite confused about the discussion about having different agent with different policies, from the rest of the paper the idea I had gotten was that the you were selecting an action that maximizes the sum of individual reward components result in different policies. I had to go back to reading the original paper (Van Seijen et al. 2017) to see that, at least by my understanding they seem to maximize each Q factor component separately. In that context it makes sense to assume there are different agents. However in your case the loss function for each head seem to take the sum of the Q values of all the components, which is different from what they were doing. Additionally, there is the question of what y^{DoubleDQN}_{c_i} is, if it is defined only in terms of the cost component c_i and chooses a next action that maximizes the Q value for just that component, then you have something that's between an optimal solver and what (Van Seijen et al. 2017). But then the question, rises about why one would want to use this method. (Van Seijen et al. 2017) motivates their method by citing the fact that, in fact the method may help apply these methods to high dimension problems and it's best to choose reward components that may use minimal number of state variables. Your problem is simple enough that you could have used an exact method, additionally your reward decomposition is more dependent on whether the reward components correspond to meaningful terms.

With that said, I still think the paper would lead to interesting discussions in the workshop and would recommend accepting it.

On a smaller note, I was a bit surprised to read that you had to teach the participants about reinforcement learning. Are the authors in some sense claiming that these methods will only be useful to people who understand RL? This would make these methods extremely narrow in its applicability and I assume it was mainly required because of the local explanation. Could one have used more intuitive terms for presenting the reward decomposition? or does the authors believe this is a fundamental limitation of the method?

---

### Meta-Review · Program_Chairs · 2022-04-30

**Recommendation:** Accept
**Confidence:** 4

**Metareview:**

The paper looks at local and global explanation methods for explaining the behavior of RL agents. The problem being addressed is very interesting and can lead to fruitful discussions in the workshop.

Nonetheless, as pointed out by the reviewers the paper has many flaws with respect to its presentation, technical description, and evaluation of the proposed approaches. One of the main concerns raised was on the paper’s underlying motivation, i.e., the choice of the specific local and global explanation techniques has not been made clear or justified by the authors. The reviewers have also pointed out issues with the paper’s clarity, technical details not being described and justified adequately, notational issues and so on.

I would recommend the authors to not only address these comments in the future iteration of the paper but to also engage and respond to the reviewers on OpenReview.

---

### Decision · Program_Chairs · 2022-04-30

Accept